# Cotton seed cultivar identification based on the fusion of spectral and textural features

Xiao Liu[1], Peng Guo[1]*, Quan Xu[2]*, Wenling Du[1]

1 College of Sciences, Shihezi University, Shihezi, China, 2 China Geological Survey Urumqi Comprehensive Survey Center on Natural Resources, Urumqi, China

* guopeng@shzu.edu.cn (PG); xuquan0928@163.com (QX)

**Data Availability Statement:** The data are available from the Open Science Framework database. https://osf.io/yxbvq/?view_only=c7aa038da26640b48127846b25d80264 DOI 10.17605/OSF.IO/YXBVQ.

## Abstract

The mixing of cotton seeds of different cultivars and qualities can lead to differences in growth conditions and make field management difficult. In particular, except for yield loss, it can also lead to inconsistent cotton quality and poor textile product quality, causing huge economic losses to farmers and the cotton processing industry. However, traditional cultivar identification methods for cotton seeds are time-consuming, labor-intensive, and cumbersome, which cannot meet the needs of modern agriculture and modern cotton processing industry. Therefore, there is an urgent need for a fast, accurate, and non-destructive method for identifying cotton seed cultivars. In this study, hyperspectral images (397.32 nm—1003.58 nm) of five cotton cultivars, namely Jinke 20, Jinke 21, Xinluzao 64, Xinluzao 74, and Zhongmiansuo 5, were captured using a Specim IQ camera, and then the average spectral information of seeds of each cultivar was used for spectral analysis, aiming to estab-lish a cotton seed cultivar identification model. Due to the presence of many obvious noises in the < 400 nm and > 1000 nm regions of the collected spectral data, spectra from 400 nm to 1000 nm were selected as the representative spectra of the seed samples. Then, various denoising techniques, including Savitzky-Golay (SG), Standard Normal Variate (SNV), and First Derivative (FD), were applied individually and in combination to improve the quality of the spectra. Additionally, a successive projections algorithm (SPA) was employed for spectral feature selection. Based on the full-band spectra, a Partial Least Squares-Discriminant Analysis (PLS-DA) model was established. Furthermore, spectral features and textural features were fused to create Random Forest (RF), Convolutional Neural Network (CNN), and Extreme Learning Machine (ELM) identification models. The results showed that: (1) The SNV-FD preprocessing method showed the optimal denoising performance. (2) SPA highlighted the near-infrared region (800–1000 nm), red region (620–700 nm), and blue-green region (420–570 nm) for identifying cotton cultivar. (3) The fusion of spectral features and textural features did not consistently improve the accuracy of all modeling strategies, suggesting the need for further research on appropriate modeling strat-egies. (4) The ELM model had the highest cotton cultivar identification accuracy, with an accuracy of 100% for the training set and 98.89% for the test set. In conclusion, this study successfully developed a highly accurate cotton seed cultivar identification model (ELM model). This study provides a new method for the rapid and non-destructive identification of

**Funding:** This research was funded by the National Natural Science Foundation of China (grant number: U2003109) and the Graduate Student Innovation Plan Project of Xinjiang Uygur Autonomous Region (grant number: 2023057).The funders had no role in study design, data collection and analysis, decision to publish, or preparation of the manuscript.

**Competing interests:** The authors have declared that no competing interests exist.

cotton seed cultivars, which will help ensure the cultivar consistency of seeds used in cotton planting, and improve the overall quality and yield of cotton.

## Introduction

China is a major producer and consumer of cotton, and the cotton industry occupies an important position in the national economy. Different cotton cultivars possess different characteristics in resistance to pests and diseases, fiber quality, yield, etc. [1–3]. The market's high quality requirements for textile products and cotton require cotton seeds to reach a certain level of purity and quality. Xinjiang is China's largest cotton production base, and there are a large number of cotton cultivars in the local seed markets. Driven by economic interests, some merchants often mix different cultivars of cotton seeds of different qualities, which can not be easily found by cotton farmers during trading and always causes great economic losses to cotton farmers and the downstream textile enterprises. Mixed seeds of different cotton cultivars can result in an inconsistency in lint quality during processing, leading to low quality of textile products and significant economic losses [4]. Additionally, the presence of cotton seeds of non-elite cultivars in the market poses a significant threat to agricultural production [5]. Therefore, accurate identification of cotton seed cultivars before cotton planting is very necessary, which is crucial for improving cotton quality, assisting cotton seed market administration, and safeguarding the interests of cotton farmers.

Traditional methods of cotton seed cultivar identification, such as field planting, protein electrophoresis, and DNA marker analysis, are complex, time-consuming, laborious, costly, and require destructive sampling, making them unsuitable for actual scenarios [6, 7]. With the rapid development of information technology, visible/near-infrared spectral feature analysis has been applied to seed cultivar identification. For example, Zhang et al. [8] developed an identification model based on near-infrared spectra (NIR) to distinguish the different cultivars of vegetable seeds. Cui et al. [9] identified a large number of maize seed cultivars based on near-infrared reflectance spectroscopy (NIRS) and stoichiometry. Visible/near-infrared (Vis/NIR) spectroscopy technology can achieve rapid and non-destructive identification of seed cultivars [10, 11]. However, because it only contains seed spectral information, the accuracy of seed cultivar identification is not high [4]. As a new technology, hyperspectral imaging technique can integrate spectra and images, that is, it not only contains rich spectral information of seeds, but also contains image information such as shape, texture, and color. This technique solves the problem that single spectral or texture information is not sufficient to distinguish the differences between different cultivars, which is helpful to improve the identification accuracy [1, 12–14]. Especially, this technology can rapidly and non-destructively analyze the internal structure and chemical composition of samples. Thus, it has gained popularity in seed inspection [15–17]. Many researchers have analyzed the relationship between spectral and image information and predicted attributes to establish models for seed cultivar identification [13–15]. For example, Sun et al. [18] used PLS-DA, support vector machine (SVM), and K-nearest neighbor model to identify black soybean seed cultivars based on spectral features, image features, and the combination of spectral and image features. Sofacles Figueredo et al. [19] successfully identified corn seed cultivars using a PLS-DA model. Zhang et al. [20] successfully identified the watermelon seeds of different cultivars using the ELM model established with spectral features. Liu et al. [21] accurately identified wheat seeds of different cultivars using a SVM model. It is important to note that different models have varying performances in the identification of seeds of different cultivars. For example, Wang et al. [22]

reported that the CNN model outperformed the SVM and KNN models in the identification of corn seeds of different cultivars.

At present, most researches on the identification of seed cultivars by hyperspectral imaging technology focus on the seeds of corn, wheat, soybean, etc. [5, 23, 24]. The performance of hyperspectral imaging technology in the identification of cotton seed cultivars is still unclear. In addition, previous hyperspectral data acquisitions require the use of a mobile platform, making the process cumbersome. Especially, the acquired hyperspectral images have to be corrected by manual computation [25, 26]. However, the portable Specim IQ handheld push-broom hyperspectral camera offers real-time data acquisition and ease of operation. Especially, the data acquired with the Specim IQ camera are already corrected, eliminating the need for additional computation, which improves efficiency. The usability of the Specim IQ camera was already demonstrated by Jan et al. [27], who compared it with the Specim V10E sensor and evaluated its measurement quality. In this study, a PLS-DA identification model was established using full spectrum data. After preprocessing the spectral data of cotton seeds with the selected optimal preprocessing method, the SPA was used to select the spectral features that were most sensitive to cotton seeds of different cultivars. Additionally, textural features were extracted using the gray-level co-occurrence matrix from the first principal component after performing principal component analysis on the hyperspectral images. Lastly, the performance of established Random Forest (RF), Convolutional Neural Network (CNN), and Extreme Learning Machine (ELM) identification models based on spectral features, texture features, or fusion of spectral features and texture features in identifying cotton seeds of different cultivars were compared. The objectives of this study were to select the optimal preprocessing method for denoising cotton seed spectral data, extract spectral features for different cultivars, and compare the performance of RF, CNN, and ELM models established based on different features. This study will provide a new method for rapid non-destructive identification of cotton seed cultivars.

## Materials and methods

### Experimental materials

Cotton seeds (cultivars Jinke 20, Jinke 21, Xinluzao 64, Xinluzao 74, and Zhongmi-ansuo 125) were obtained from a seed company (Jinfenghe, Shihezi, China) (60 seeds per cultivar). The variations in shape, size, and skin roughness among the seed samples were minimized [4], to avoid any potential errors in the spectral data caused by incon-sistency in the seeds.

### Hyperspectral image acquisition

In this study, a Specim IQ hyperspectral camera (Oulu, Finland) was used to acquire hyperspectral images of cotton seeds. The camera has a wavelength range of 400–1000 nm and a spectral resolution of 7 nm. It captures images with a pixel count of 512 per line and provides 204 spectral points within the wavelength range.

The Custom mode of the Specim IQ camera was used for image acquisition. A white board was used as a reference. Green materials with distinct color difference and different absorbance from the epidermis of cotton seeds were selected as the background for indoor image acquisition. The cotton seeds were placed individually for hyperspectral image acquisition (Fig 1).

The obtained hyperspectral images were synthesized into pseudo-colors using three bands: 449.35, 548.55, and 598.60 nm. These images showed the spectral features of the cotton seeds and were used for further analysis of differences between cultivars.

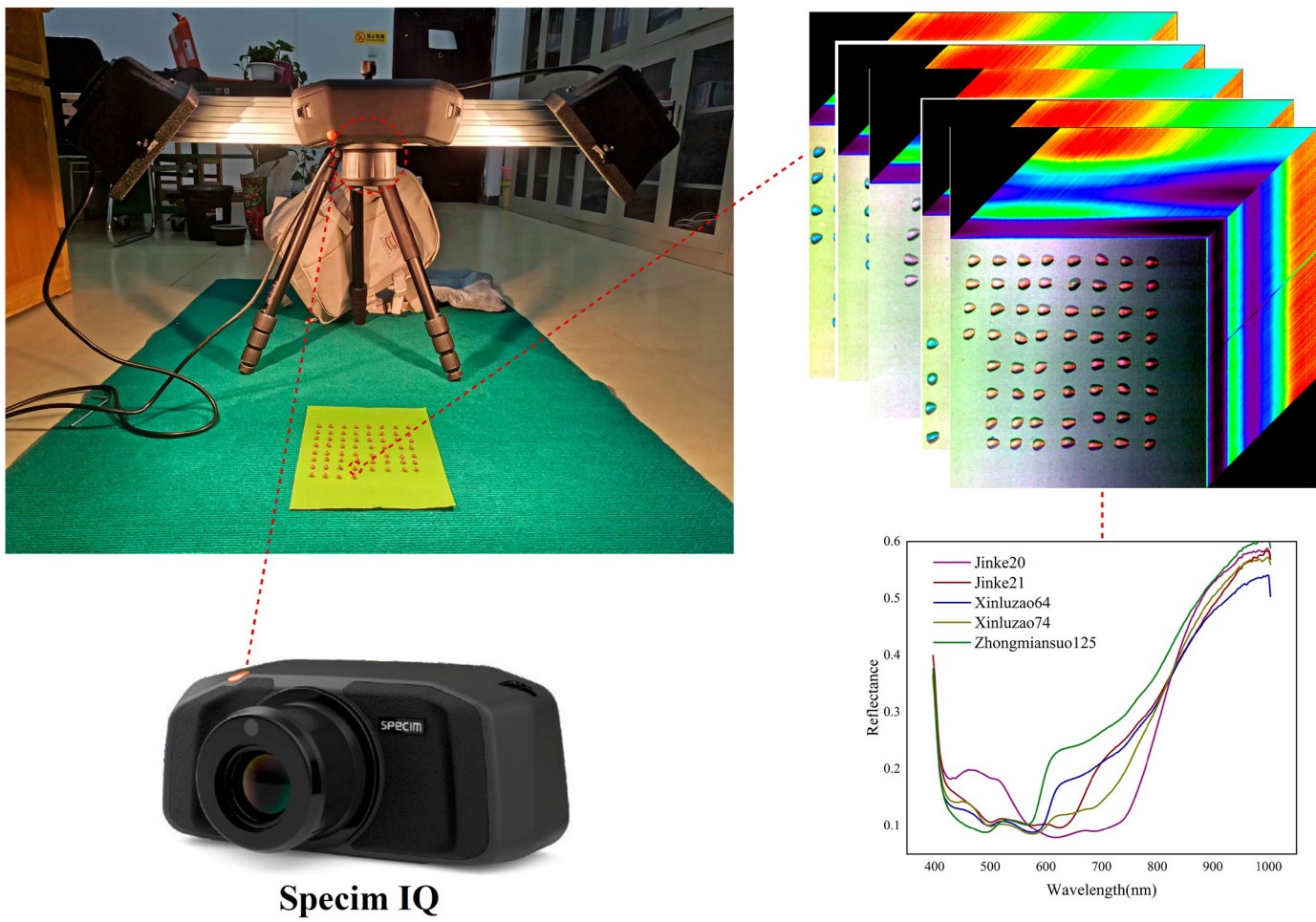

**Fig 1. Hyperspectral image acquisition system.**

### Data sets

The ENVI 5.6 software was used to extract the spectral data of cotton seeds, with each cotton seeds as a regions of interest (ROI).

To obtain th**e** average spectrum of each seed, the average of the spectra of all pixels within the ROI was calculated. This yielded 300 spectral data for each cotton cultivar. These data were saved in a matrix with a dimensions of 300 × 204 (raw spectral dataset). The raw spectral dataset was processed and analyzed, to investigate the differences between different cultivars and extract their spectral features.

The data of 42 seeds of each cultivar were used as the training set, and the data of remaining 18 seeds were used as the test set (7: 3) (Table 1). The value 1, 2, 3, 4, and were assigned to the data of Jinke 20, Jinke 21, Xinluzao 64, Xinluzao 74, and Zhongmiansuo 125, respectively.

**Table 1. Division of training set and testing set.**

|  | Jinke20 | Jinke21 | Xinluzao64 | Xinluzao74 | Zhongmiansuo125 |
|---|---|---|---|---|---|
| **Class label** | 1 | 2 | 3 | 4 | 5 |
| **Training set** | 42 | 42 | 42 | 42 | 42 |
| **Testing set** | 18 | 18 | 18 | 18 | 18 |

## Data analysis

### Spectral preprocessing

Background, uneven light source, etc. caused interference to hyperspectral image acquisition, yielding noises in the raw spectral. To reduce the influence of these interferences and extract spectral features, three denoising algorithms, Savitzky-Golay smoothing (SG), first-order derivative (FD), standard normal transform (SNV), and their combinations (SG-SNV, SG-FD, SNV-FD, and SG-SNV-FD) were used to preprocess the raw spectra. Besides, the denoising performances were compared.

The SG smoothing is a smoothing method based on local polynomial fitting [28]. It smooths spectral curve by fitting a local polynomial to reduce the influence of high-frequency noise. The choice of smoothing points directly affects the smoothing performance [29, 30]. In this study, the SG quadratic polynomial 7-point smoothing was chosen to preprocess the spectral data (Eq (1)).

$$x_{a,smooth} = \frac{1}{H} \sum_{i=-\omega}^{\omega} x_{a+i} h_i \tag{1}$$

Where $x_{a,smooth}$ is the reflectance after SG preprocessing, $H$ is the normalization factor, and $\omega$ is the window width 1/2.

The first derivative highlights the edge features of the spectrum and suppresses low-frequency noise. By deriving the spectral data, the negative effects caused by spectral baseline drift can be eliminated [31, 32].

$$D(\lambda_I) = \boldsymbol{R}(\lambda_{i+1}) - \boldsymbol{R}(\lambda_{i-1})/2\Delta\lambda \tag{2}$$

where $\boldsymbol{D}(\lambda_\mathbf{i})$ represents the first-order derivative reflectance at wavelength i, and $\boldsymbol{R}(\lambda_\mathbf{i+1})$ **and** $\boldsymbol{R}(\lambda_\mathbf{i-1})$ represent the raw spectral reflectance at wavelengths i+1 and i-1, respectively.

The Standard Normal Variate (SNV) Transform can eliminate the impacts of solid particle size, surface scattering, and brightness differences on the spectral data [33]. The spectral processing by SNV algorithm is based on the rows of spectral matrix (Eqs (3) and (4)).

$$x_{SNV} = \frac{x - \omega_1}{\sqrt{\frac{\sum_{k=1}^{m}(x_k-\omega_1)^2}{(m-1)}}} \tag{3}$$

$$\omega_1 = \frac{\sum_{k=1}^{m} x_k}{m} \tag{4}$$

where $\boldsymbol{x_{SNV}}$ is the reflectance after SNV preprocessing, x is the average reflectance, and m is the number of wavelengths.

### Feature selection

Feature selection was conducted to eliminate redundant and collinear information in full-spectrum data. These redundancies not only hinder the extraction of spectral features, but also increase computational complexity. Therefore, the SPA, a forward variable selection algorithm that minimizes collinearity, was employed to eliminate redundant information within the raw spectra and extract spectral features [34–36]. The spectral data preprocessed by the optimal preprocessing method and category assignments were used as inputs. Through continuous iterations, the features were selected. Multiple linear regression analysis was performed after each iteration. The algorithm ran until the root mean square error (RMSE) of the Testing set

stabilized and reached the minimum value. When the RMSE of the Testing set tended to stabilize and reached a minimum, the output was the optimal number of variables, spectral features and their contributions. At this point, the optimal number of features was selected.

## Textural feature extraction

Textural features show the structural features of the object surface in hyperspectral images. These features can provide valuable information for distinguishing cotton seed cultivars. In this study, the gray-level co-occurrence matrix including 14 types of textural features was used to extract textural features [37]. The gray-level co-occurrence matrix provides information about the spatial relationships between pixels, which is useful for texture analysis [38].

Firstly, the first principal component after principal component analysis (PCA) was selected. This component represents the most significant variation in the data and helps reduce dimensionality [39]. Then, eight commonly used textural features were selected from the gray-level co-occurrence matrix. These features include mean, variance, homogeneity, contrast, dissimilarity, entropy, second-order moments, and correlation.

The size of the moving window used in textural feature extraction is crucial [40]. In this study, the identification performance using window sizes of $3 \times 3$, $5 \times 5$, $7 \times 7$, and $9 \times 9$ were compared. It was found that $3 \times 3$ was the optimal window size.

The extracted useful textural features were used in modeling. These textural features provide additional information beyond spectral data and can enhance the accuracy and reliability of the identification model.

## Machine learning

A PLS-DA (Partial Least Squares Discriminant Analysis) model based on full-band spectral data of cotton seeds of the five cultivars was established to evaluate the influence of different preprocessing methods on the accuracy of cotton seed cultivar identification model. The PLS-DA algorithm is a supervised regression model-based PLS multivariate statistical analysis [41, 42]. It utilizes the spectral data to construct a identification model for identification purposes. The study employed the RF algorithm, known for its ability to handle high-dimensional features and fast training speed, for modeling [43]. RF constructs multiple decision trees by randomly selecting features, and classifies them through a voting mechanism, which helps mitigate overfitting [44].

Additionally, the study used the CNN model, a deep learning algorithm capable of distinguishing spectral and image information for modeling. CNN achieves accurate detection and prediction through hierarchical-layer stacking and specially designed network structures [45, 46]. In this study, the CNN model structure which is modified on the basis of the model structure of Wang et al. [22] consisted of 12 layers, including an input layer, two convolutional layers, two normalization layers, two activation layers, two pooling layers, two fully connected layers, and one output layer. The size of the convolution kernel was 2*1, and the activation function was RELU. To avoid overfitting, L2 regularization was incorporated in both convolutional layers, and the coefficient was set to 0.0001. Lastly, the ELM algorithm was employed for modeling [47]. ELM randomly initializes the hidden-layer weights and uses simple linear equations to obtain the output layer weights. Compared with traditional neural network models, ELM has faster learning speed and better generalization ability [48].

The performance of the cotton seed cultivar identification models established by combining selected spectral features and textural features based on these three algorithms were compared.

## Model evaluation

In this study, the Accuracy (the proportion of correctly identified cotton seed samples to the total samples) was used to evaluate the model. The higher the Accuracy, the higher identification accuracy of the model.

## Results

### Selection of optimal spectral pre-processing methods

Raw spectral data consisted of 204spectral points (wavelength range: 397–1004 nm). The spectral reflectance of the five cotton cultivars mostly overlapped across the entire spectral range (Fig 2(a)), except for 400–700 nm and 800–900 nm (Fig 2(b)). Notably, the differences in spectral reflectance between Jinke 20 and the other four cultivars were most apparent. To facilitate feature selection from the raw data, noises at the front and back ends of the spectra were eliminated. Finally, the spectra between 400 and 1000 nm were preprocessed using SG, SNV, FD, SG-SNV, SG-FD, SNV-FD, and SG-SNV-FD.

The SG smoothing yielded smooth curves and eliminated jagged noises across the entire spectral range (Fig 3(a)). SNV transform reduced intra-cultivar spectral differences, but increased inter-cultivar differences (Fig 3(b)). The first-order derivative highlighted the slope changes in the raw spectral data, and made the spectral peaks' positions and shapes more prominent, highlighting the spectral features (Fig 3c). In summary, the preprocessings significantly eliminated noises in the raw spectral data, yielding a more reliable database for subsequent feature extraction.

In this study, the PLS-DA of the training and Testing sets of full-band spectra that were preprocessed was performed. Table 2 shows the results of the PLS-DA results for each preprocessing method. The optimal results were obtained for the spectra preprocessed using the SNV-FD method. The accuracy of both the training set and the Testing set was 1. It was worth noting that while the SG smoothing eliminated noises from the spectra, it also eliminated some valid information, resulting in the poorest PLS-DA results.

### Feature selection

In this study, fifteen variables were selected as spectral features (569.12 nm, 637.08 nm, 472.59 nm, 501.72 nm, 960.40 nm, 954.24 nm, 868.55 nm, 905.18 nm, 966.55 nm, 963.47 nm, 917.42 nm, 874.64 nm, 975.79 nm, 920.48 nm, and 420.40 nm) (Fig 4a), and the contribution of each was calculated.

Among these wavelengths, the absorption peaks in the near-infrared (NIR) region are primarily attributed to the water content of the cotton seeds, especially the presence of the O—H group. The absorption peaks near the red region are related to the stretching vibration of the C—H chemical bond in the samples [49]. These bands near the red region are crucial for capturing the differences in chemical composition and physical state among different seeds [50].

To validate the selected spectral features, the positional distribution of the spectral features was analyzed using the data from the first sample of Jinke 20. The results showed that these spectral features were predominantly distributed in the near-infrared (800–1000 nm), red (620–700 nm), and blue-green (420–570 nm) regions (Fig 4b).

Overall, the selected spectral features provided valuable information for identifying different cotton seed samples by chemical composition and physical state.

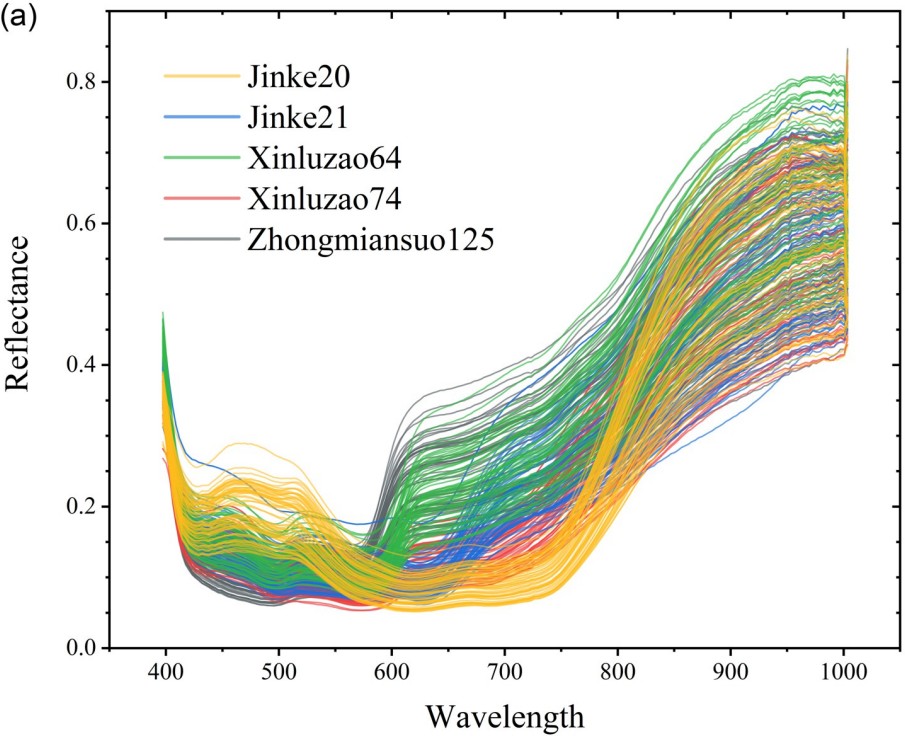

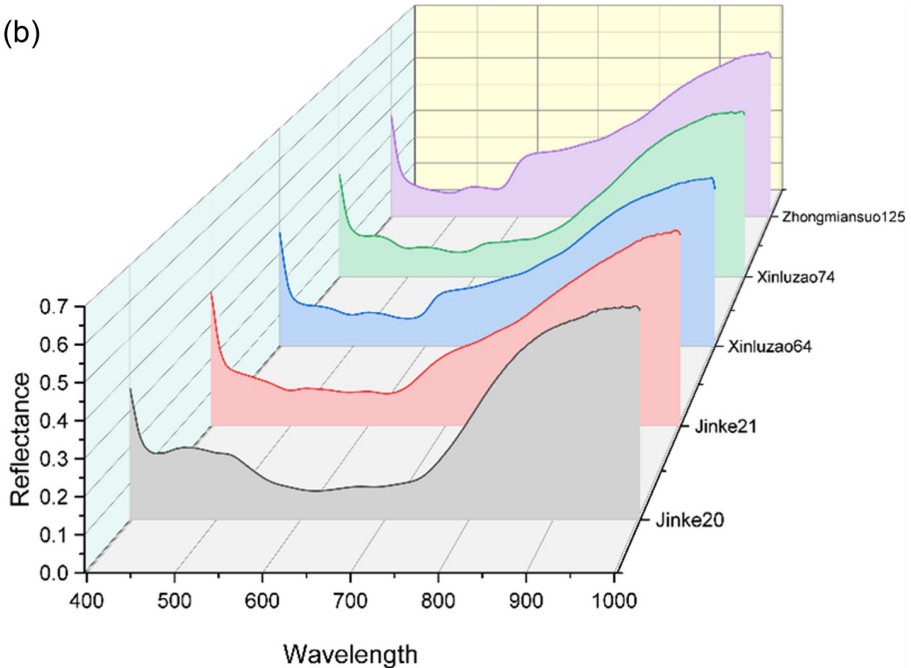

**Fig 2. Average spectrum of single seeds (a) and average spectrum of single cultivars (b).**

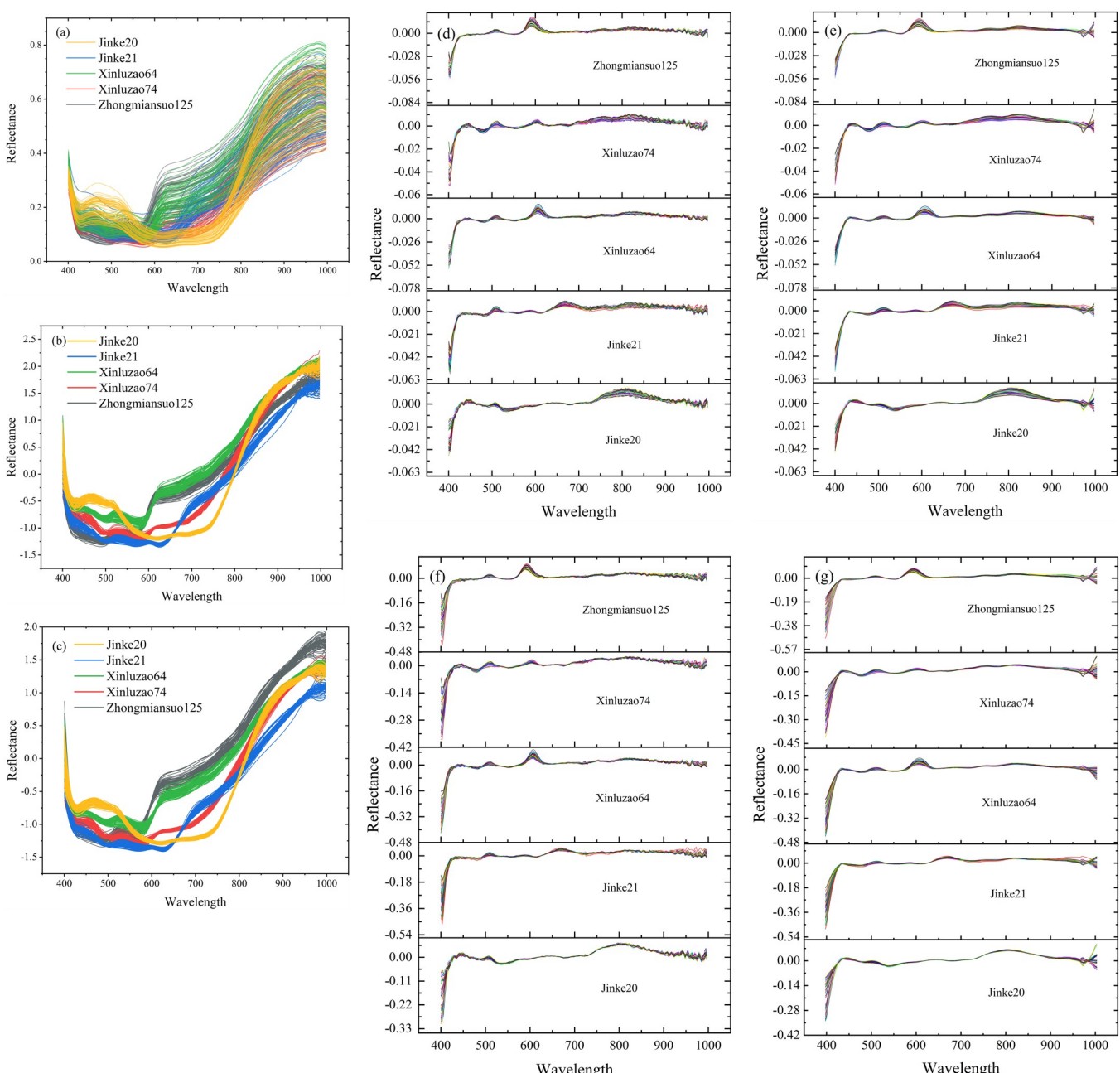

**Fig 3. Spectral reflectance curves after preprocessing with SG smoothing (a), SNV (Standard normal variate) (b), SG—SNV (c), FD (First derivative) (d), SG—FD (e), SNV—FD (f), and SG—SNV–FD (g).**

## Cotton seed cultivar identification based on spectral features

In this study, the 15 spectral features extracted by SPA after SNV-FD preprocessing were used for RF, CNN, and ELM modeling. The model identification results based on the training set and test set are presented (Fig 5). All three models achieved good identification results on both the training and test sets. The SNV-FD-SPA-ELM model performed the best, achieving 100% accuracy on the training set and 98.89% accuracy on the test set. The SNV-FD-SPA-RF and SNV-FD-SPA-CNN models performed worse.

**Table 2. Identification results of PLS-DA models based on different preprocessing methods.**

| Preprocessing | Training set | Testing set |
|---|---|---|
| | Accuracy | Accuracy |
| Raw spectrum | 0.9907 | 0.9672 |
| SG | 0.9766 | 0.9444 |
| SNV | 0.9952 | 1.0000 |
| FD | 1.0000 | 0.9833 |
| SG-SNV | 0.9904 | 1.0000 |
| SG-FD | 1.0000 | 0.9825 |
| SNV-FD | 1.0000 | 1.0000 |
| SG-SNV-FD | 0.9862 | 1.0000 |

These results indicate that the extracted spectral features can be used for identifying different cultivars of cotton seeds. The SNV-FD preprocessing method effectively eliminated noises in the spectrum, the SPA extracted the most representative wavelengths, and the ELM modeling yielded optimal identification results.

## Cotton seed identification based on textural features

The analysis results of the textural features of the five cultivar seeds (Table 3) showed that the Mean, Variance, and Contrast features exhibited significant differences between cultivars, making them useful for identifying cotton seed cultivars. The Homogeneity feature of Xinluzao 74 was obviously different from that of the other four cultivars. The Dissimilarity feature was effective in identifying Jinke 21 and Xinluzao 74 from the other three cultivars. Although Entropy, Second Moment, and Correlation features did not exhibit obvious differences, they still had subtle variations that could contribute to identifying the five cotton cultivars to some extent.

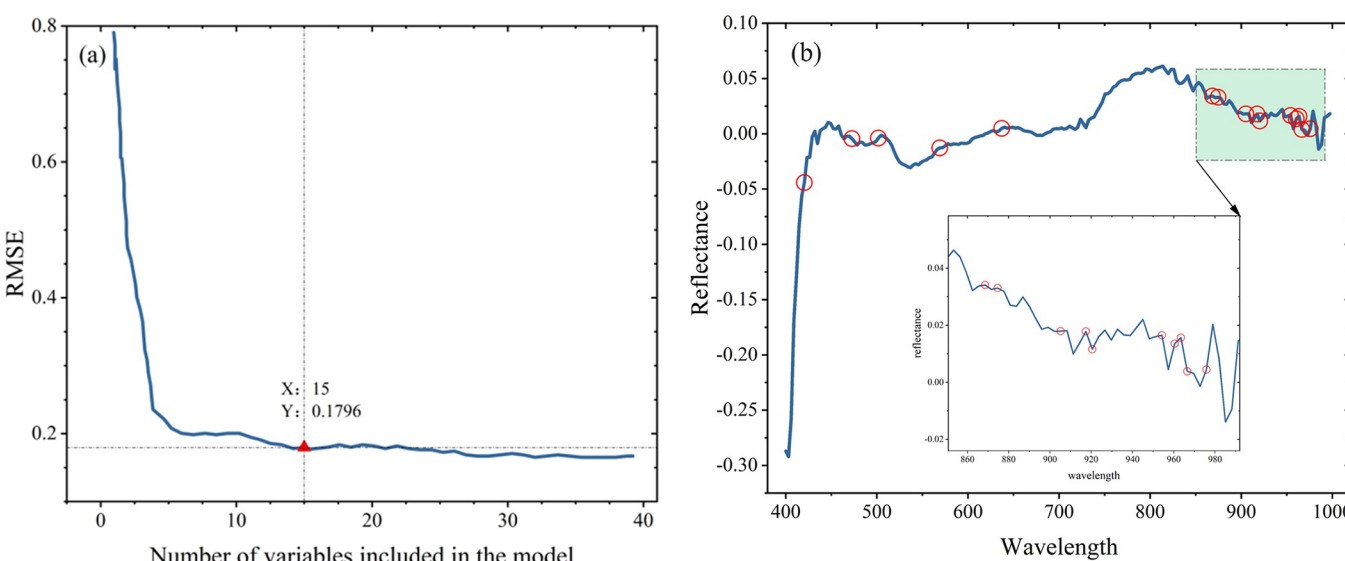

**Fig 4. Number of wavelengths selected by successive projections algorithm (SPA) (a) and the wavelengths selected by successive projections algorithm (SPA) (b).**

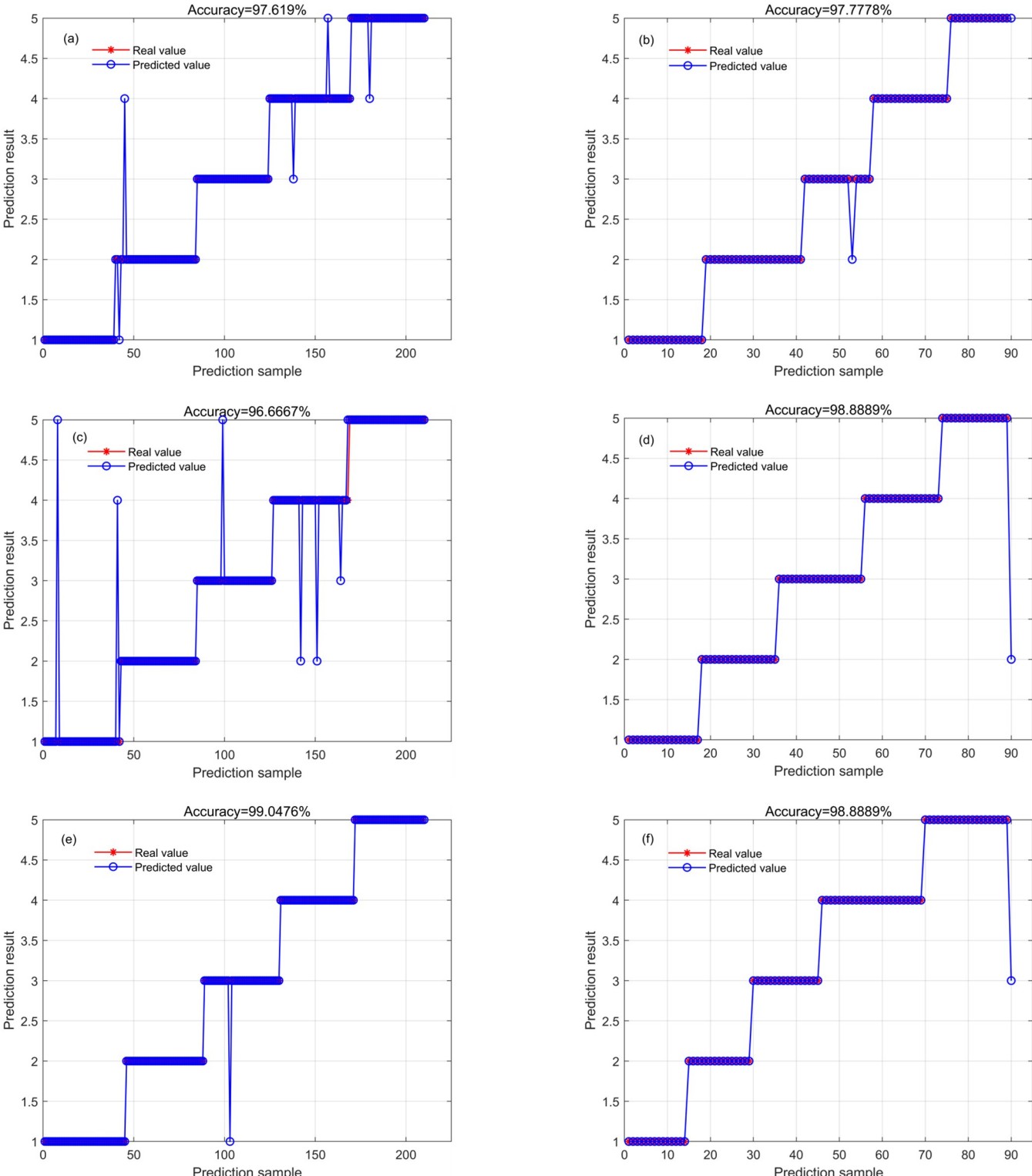

**Fig 5. The identification results of RF (a, training set; b, test set), CNN (c, training set; d, test set), and ELM (e, training set; f, test set) models based on spectral features extracted by SPA.**

**Table 3. Mean values of textural features of each cotton cultivar.**

| Type | Mean | Variance | Homogeneity | Contrast | Dissimilarity | Entropy | Second Moment | Correlation |
|---|---|---|---|---|---|---|---|---|
| Jinke 20 | 19.87153 | 15.09690 | 0.290950 | 31.91145 | 3.771268 | 2.006423 | 0.146926 | 0.602545 |
| Jinke 21 | 20.68196 | 17.92805 | 0.268794 | 36.43023 | 4.117625 | 2.047019 | 0.138678 | 0.629179 |
| Xinluzao 64 | 42.76330 | 8.522156 | 0.284408 | 17.69716 | 3.198072 | 2.002946 | 0.212020 | 0.651594 |
| Xinluzao 74 | 43.54759 | 13.70734 | 0.312784 | 25.77343 | 3.412763 | 2.00116 | 0.147801 | 0.603902 |
| Xinluzao 125 | 39.79408 | 15.74696 | 0.252170 | 33.01843 | 4.107894 | 2.05097 | 0.137788 | 0.622886 |

Based on these textural feature analysis results, the RF, CNN, and ELM models were established. Among these models, the ELM model demonstrated the best identification performance (Fig 6). However, compared with the identification based on spectral features, the identification performance based on textural features was poorer, with an accuracy of 80.48% for the training set and 71.11% for the test set. The RF and CNN models realized the highest identification accuracy for Xinluzao 64, with the number of misclassified samples not exceeding three for both training and test sets. This may be due to the fact that the Second Moment and Correlation features of Xinluzao 64 were obviously different from those of the other four cultivars. It should be noted that compared with Xinluzao 64, Jinke 20 and Jinke 21 were more likely to be mis-classified, as were Xinluzao 74 and Zhongmiansuo 125.

## Identification of cotton seed cultivars by combining spectral and textural features

Fig 7 presents the results of cotton seed **cultivar** identification by combining spectral and textural features. Comparing with the results (Figs 5 and 6), it is evident that the accuracy of both RF and ELM models are higher. After combining spectral and textural features, the ELM model had the highest accuracy (100% for the training set and 98.89% for the test set) among the three models. The CNN model showed a significant increase in accuracy compared with the CNN model based on textural features, but showed a decrease in accuracy (19.52% and 26.67% for the training and testing sets, respectively) compared with the CNN model based on spectral features.

## Discussion

### Effects of different preprocessings

Previous studies have explored the selection of pre-processing methods for identification tasks. For instance, a study on cotton seed cultivar identification found that the combination of SG (Savitzky-Golay) smoothing (with a seven-point quadratic filter) and normalization yielded the best results [4]. Sharma et al. [41] confirmed that the model based on the SG2 pre-processing obtained the highest wheat seed cultivar identification accuracy. Additionally, a study on the identification of frost-damaged rice seeds using hyperspectral imaging and a deep forest model found that the Multiplicative Scatter Correction pre-processing was the most effective in increasing identification accuracy [51]. In this study, various spectral pre-processing methods were compared to select the optimal one for pre-processing the raw spectra. The performance of different pre-processing methods was evaluated by building a PLS-DA model using the pre-processed full-band spectral data. It was found that the combination of SNV and FD was the most effective in eliminating noises from the spectra of cotton seeds and enhancing the accuracy of the model. Compared with the SG smoothing combined with normalization spectral preprocessing method used by Huang et al. [4] in the study of the cotton seed cultivar

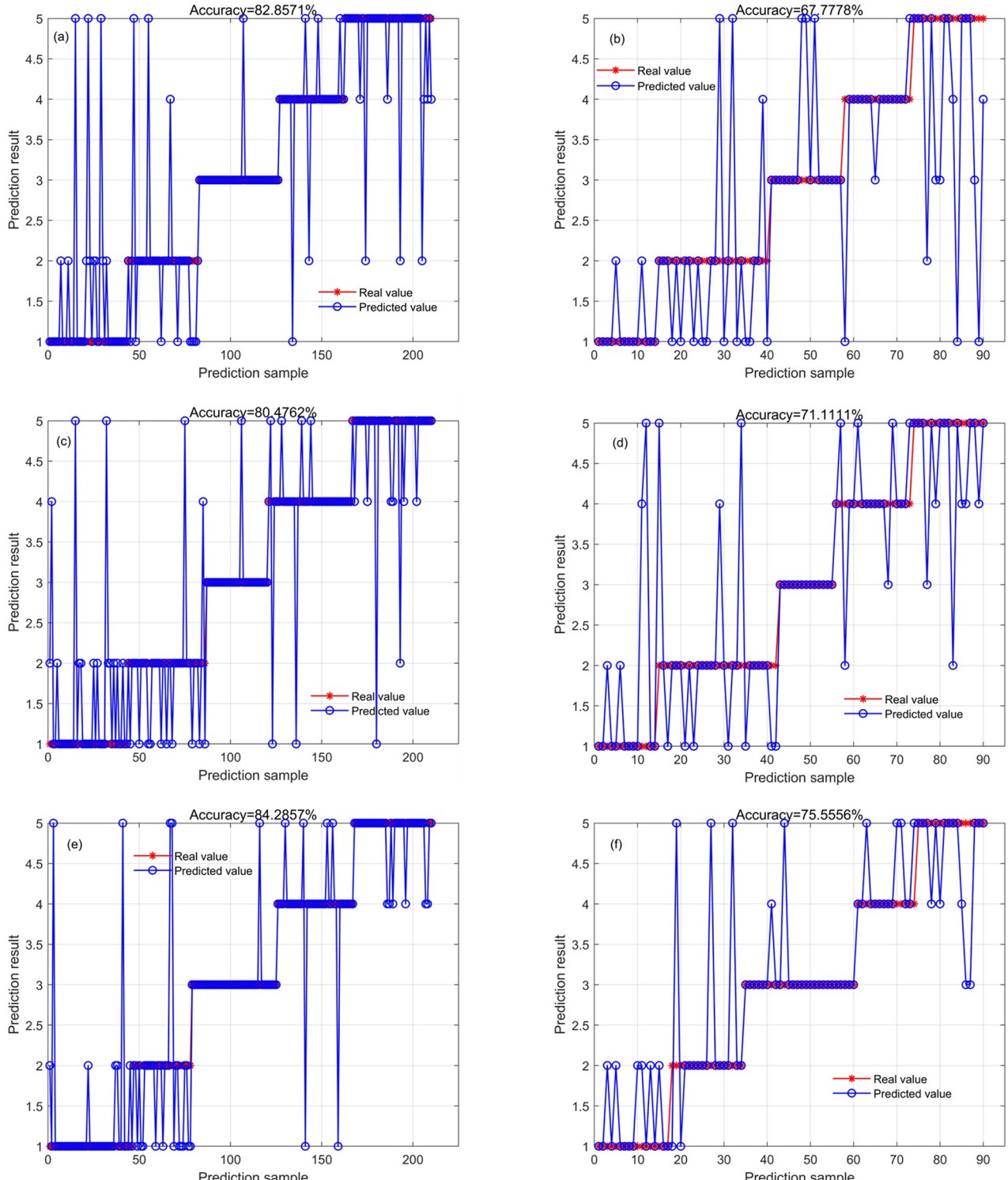

**Fig 6. The identification results of the RF (a, training set; b, test set), CNN (c, training set; d, test set), and ELM (e, training set; f, test set) models based on textural features extracted by gray-level co-occurrence matrix.**

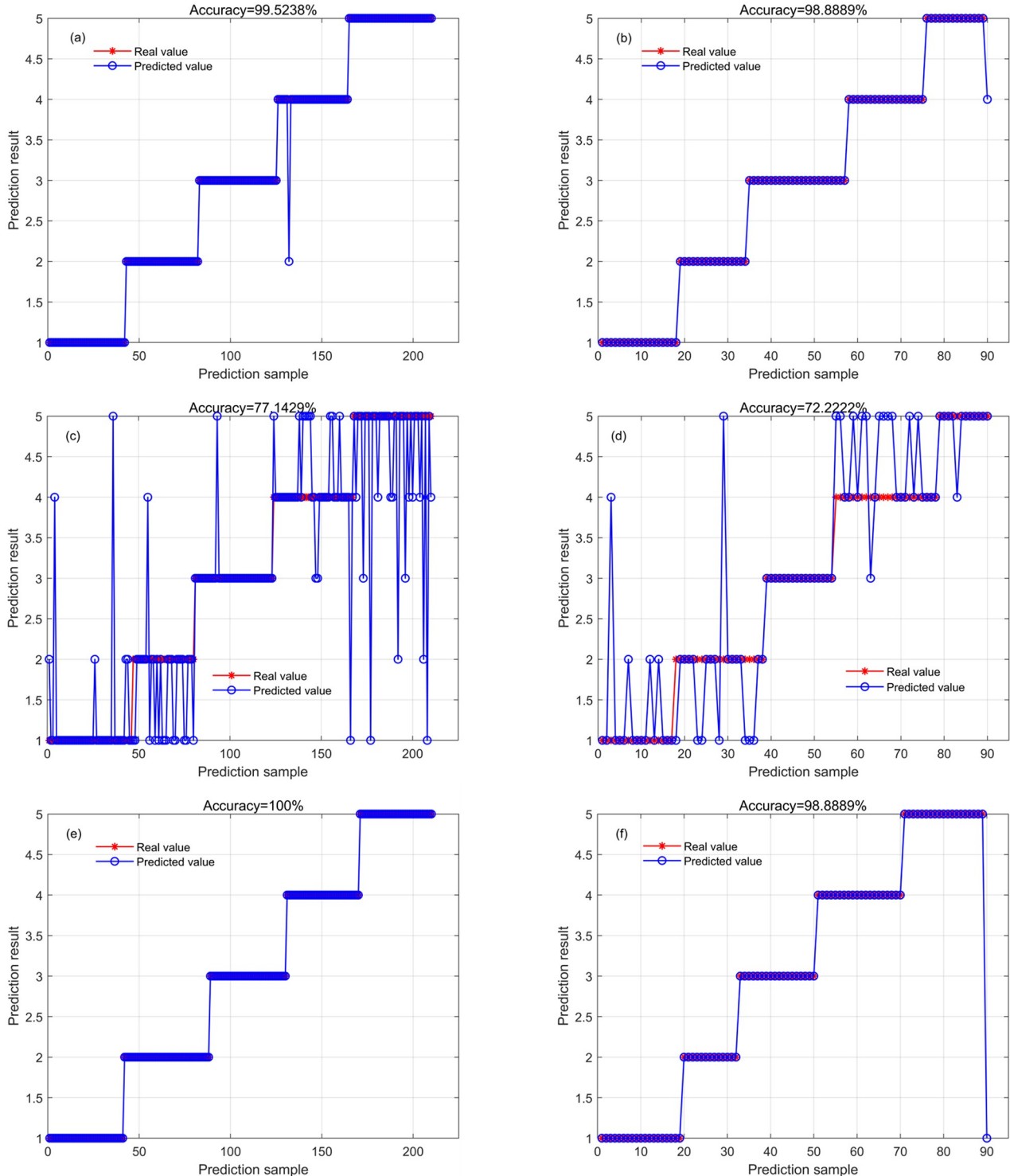

**Fig 7. The identification results of the RF (a, training set; b, test set), CNN (c, training set; d, test set), and ELM (e, training set; f, test set) models established by combining spectral and textural features.**

identification based on hyperspectral image technology, the SNV-FD preprocessing method used in this study can effectively reduce the noise of cotton seed spectral data. Especially, the identification accuracy of the PLS-DA model based on the spectral data preprocessed by SNV-FD in this study is 4% (training set) and 6% (test set) higher than that of the PLS-DA model based on the spectral data preprocessed by SG smoothing combined with normalization [4]. These findings highlight that different pre-processing methods can yield different outcomes depending on the specific context and object characteristics. Therefore, when selecting the optimal pre-processing method, it is crucial to consider the specific environment, object characteristics, etc., and conduct evaluations. This can ensure the high quality and usability of spectral data for the given task.

## Spectral variability analysis

Feature selection using the SPA could effectively reduce the redundant information, improving the processing efficiency of spectral data and the accuracy of the identification model [34–36]. Therefore, in this study, the SPA was employed for spectral feature selection, and the importance of different wavelengths in identifying cotton seed cultivars was analyzed. It was found that the top 15 wavelengths by contribution were predominantly in the near-infrared (800–1000 nm), red (620–700 nm), and blue-green (420–570 nm) regions. This indicates that the spectral features in the blue-green, red, and red-edge regions are crucial in differentiating cotton seed cultivars. The spectral features in the blue-green region can reflect the chlorophyll content and growth status of cotton plants [4]. The spectral features in the red region can reflect the leaf thickness and chlorophyll content of cotton plants [52]. The spectral features in the red-edge region can reflect the leaf structure and growth status of cotton plants [53, 54]. However, the chlorophyll content, leaf thickness, leaf structure, and growth status of different cotton cultivars vary greatly. Hence, the blue-green red, and red-edge bands can be used to identify different cotton seed cultivars. By comprehensively using the spectral features of the blue-green, red, and red-edge regions, accurate identifying of different cotton seed cultivars can be realized.

## Identification performance based on the combination of spectral and textural features

In this study, to validate the effectiveness of textural features for the identification of cotton seed cultivars, spectral features were combined with eight textural features selected based on the gray scale covariance matrix for identification. The results showed that the inclusion of textural features significantly improved the identification accuracy of the RF and ELM models, that is, the overall identification accuracy of the models established using spectral and textural features was higher than that of the models based solely on full-spectrum data, spectral features, or textural features (Figs 5–7). However, the accuracy of the CNN model decreased after the inclusion of textural features (Figs 5–7). This indicates that the increase in the number of features does not necessarily increase the identification accuracy. This may be due to the fact that the spectral features play a dominant role in identification, and the CNN model may become overfitted or insufficiently trained when textural features are added, leading to a decrease in identification accuracy. Therefore, when constructing a model for cotton seed cultivars identification, the selection of appropriate features is very necessary. Additionally, it is crucial to combine selected features with a suitable modeling strategy for further optimization. This can help identify the most effective feature sets and the optimal modeling strategy to achieve the highest accuracy in cotton seed cultivars identification.

In this study, various preprocessing methods were combined with RF, CNN, and ELM algorithms for cotton seed cultivar identification. The highest accuracy (100%) was achieved through combining the SNV-FD preprocessing, the fusion of spectral and textural features, and the ELM modeling strategy. This is similar to the watermelon seed cultivar identification accuracy using the near-infrared hyperspectral image technology by Zhang et al. [20]. However, there are still some limitations in this study. In this study, seeds were separated for hyperspectral image acquisition. However, in actual cases, the seeds are stacked. Whether the model could still achieve good results in the case of stacking needs to be tested in future study. In addition, in this study, five cotton cultivars widely planted in Xinjiang, China were studied. However, in the cotton seed trade market, there are many cultivars available. So, in the follow-up research, more cotton cultivars will be included in the study to improve the stability and applicability of the model.

## Conclusions

In this study, the raw spectral data was preprocessed by various spectral preprocessing methods before feature selection using the successive projections algorithm. After that, the spectral and textural features were fused to establish RF, CNN, and ELM identification models. The main conclusions were as follows:

The SNV-FD preprocessing method effectively eliminated noises and highlighted spectral features. Based on the selected spectral features, the PLS-DA model achieved the highest accuracy (the accuracy, $R^2$, and $Q^2$ were 1.0000, 0.9883, and 0.9868, respectively for the training set, and 1.0000, 0.9900, and 0.9818, respectively for the test set).

The importance of different spectral regions in distinguishing cotton seed cultivars varied. The spectral features selected from the near-infrared (800–1000 nm), red (620–700 nm), and blue-green (420–570 nm) regions by the SPA algorithm were found to be the most effective for identifying cotton seed cultivars. Utilizing the information from these bands improved the cotton seed cultivar identification accuracy.

The inclusion of textural features did not universally improve the accuracy of all models. The identification accuracy of RF and ELM models was significantly improved, while that of the CNN models was decreased. This may be attributed to overfitting or insufficient training of the CNN model after fusing spectral and textural features. Therefore, selecting appropriate modeling strategy is crucial to enhance identification accuracy in practical applications.

The ELM model showed the highest cotton seed cultivar identification accuracy (100% for the training set and 98.89% for the test set). Therefore, the ELM model has strong generalization ability and can be applied in practical productions.

Overall, the study highlights the importance of the selection of appropriate preprocessing methods, feature selection techniques, and modeling strategies in cotton seed cultivar identification, and provides valuable reference for practical applications.

## Author Contributions

**Data curation:** Wenling Du.

**Supervision:** Peng Guo, Quan Xu.

**Writing – original draft:** Xiao Liu.

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
