## [Decision Letter · Decision Letter 0]

26 Dec 2023

PONE-D-23-36209Cotton Seed Cultivar Identification Based on the Fusion of Spectral and Textural FeaturesPLOS ONE

Dear Dr. Liu,

Thank you for submitting your manuscript to PLOS ONE. After careful consideration, we feel that it has merit but does not fully meet PLOS ONE’s publication criteria as it currently stands. Therefore, we invite you to submit a revised version of the manuscript that addresses the points raised during the review process.

We look forward to receiving your revised manuscript.

Kind regards,

Narendra Khatri, Ph.D.

Academic Editor

PLOS ONE

Journal Requirements:

This research was funded by the National Natural Science Foundation of China (grant number: U2003109) and the Graduate Student Innovation Plan Project of Xinjiang Uygur Autonomous Region (grant number: 2023057).

5. In the online submission form, you indicated that The data that support the findings of this study are available on request from the corresponding author, upon reasonable request.

Reviewers' comments:

Reviewer's Responses to Questions

**Comments to the Author**

1. Is the manuscript technically sound, and do the data support the conclusions?

Reviewer #1: Yes

Reviewer #2: Partly

2. Has the statistical analysis been performed appropriately and rigorously? 

Reviewer #1: Yes

Reviewer #2: Yes

3. Have the authors made all data underlying the findings in their manuscript fully available?

Reviewer #1: Yes

Reviewer #2: Yes

4. Is the manuscript presented in an intelligible fashion and written in standard English?

Reviewer #1: No

Reviewer #2: Yes

5. Review Comments to the Author

Reviewer #1: The manuscript is well written with some new findings.

There are few things that need to be improved-

a through proof read is required for grammatical and typing errors.

Elaborate Figure 6.

What are the practical applications of your research findings and mention the constraints associated with it (if any )??

What are the limitations of your work?

Reviewer #2: The paper presents an experimental study for cotton seed identification using hyperspectral images analysis. The authors have tried various methods at each stage of the classification task, starting from the preprocessing stage of noise removal, feature selection and model training and evaluation. Overall, the paper is well-written with clearly explained steps.

However, I would like to suggest the following points which the authors can incorporate in their manuscript after revision.

- The contributions need to be highlighted clearly in the introduction, with clear distinction from the state-of-the-art.

- The literature survey is not thorough. It should highlight clear differences and drawbacks that this work overcomes. A citation of Sun et al. [17] have a similar methodology and workflow. The authors should clearly mention how this work is different from theirs, or is it just the application to cotton seeds that is different.

- The overall results are not being compared with state-of-the art methods. The authors have mentioned different preprocessing/filtering methods used in different literature review papers, but the final results need to be compared. If they are better, then why they are better.

- Training and evaluation methodology and sample size: The authors have taken a relatively low sample size with 60 samples per class. Moreover, the authors have mentioned that the samples were hand-picked to minimize the intra-class variations. This has resulted in nearly 100% accuracy, which ideally, should not be presented. A perfect accuracy generally, suggests that the sample size needs to be increased.

- Model details can be mentioned. Like when using CNNs, what where the layers and sizes that were considered and why they were considered.

- The partition of data can be reconsidered with clear reasoning. The authors have taken training and validation sets, but have not taken a test set. Is the validation set being used as the test set? If not, then what (hyper)parameters are being modified using the validation set. At what stages of pre-processing / modeling / dimensionality reduction, are the training set being considered separately, or it is that the validation and training sets are being used together. Is there a dependence of test / validation set on the training data during the dimensionality reduction step. This point seems unclear.

- The authors may point out the problems of working with raw spectral data. They can also mention the reasons why the recent state of the art detection/classification models cannot be applied for this task and how the spectral data poses computational challenges?

- Fig. 2 (a) and (b) labels are missing.

- The authors may look into improving the visualizations of spectral data.

6. PLOS authors have the option to publish the peer review history of their article (what does this mean?). If published, this will include your full peer review and any attached files.

Reviewer #1: No

Reviewer #2: No

---

## [Author Response · Author response to Decision Letter 0]

31 Jan 2024

Dear editor and reviewers, 

Many thanks for giving us this opportunity to improve our manuscript. We sincerely thank the reviewers for taking the time to review our research article, and greatly appreciate the comments and suggestions on our research work and article content. We have made corresponding modifications based on these comments. We make every effort to ensure that all changes are accurate and that every modification is valid. We hope that the revised version will meet the publication standard.

Reviewer #1: The manuscript is well written with some new findings.

There are few things that need to be improved-

a through proof read is required for grammatical and typing errors.

Response: Many thanks for your review and comments. We have thoroughly checked the manuscript, and improved the language with the help of a native English speaker. 

Elaborate Figure 6.

Response: Thanks for your suggestion. We have added a detailed description for Figure 6 in the manuscript. Please see lines 325 - 335. 

" Based on these textural feature analysis results, the RF, CNN, and ELM models were established. Among these models, the ELM model demonstrated the best identification performance (Fig. 6). However, compared with the identification based on spectral features, the identification performance based on textural features was poorer, with an accuracy of 80.48% for the training set and 71.11% for the test set. The RF and CNN models realized the highest identification accuracy for Xinluzao 64, with the number of misclassified samples not exceeding three for both training and test sets. This may be due to the fact that the Second Moment and Correlation features of Xinluzao 64 were obviously different from those of the other four cultivars. It should be noted that compared with Xinluzao 64, Jinke 20 and Jinke 21 were more likely to be mis-classified, as were Xinluzao 74 and Zhongmiansuo 125."

What are the practical applications of your research findings and mention the constraints associated with it (if any )??

Response: The aim of this study was to establish a model for rapid, non-destructive, and accurate identification of cotton seed varieties, to assist seed trade and safeguard the interests of cotton farmers. The practical application of this study has been added to the first paragraph of the Introduction. Please see lines 45 - 52 and 55 - 58.

"The market's high quality requirements for textile products and cotton require cotton seeds to reach a certain level of purity and quality. Xinjiang is China's largest cotton production base, and there are a large number of cotton varieties in the local seed markets. Driven by economic interests, some merchants often mix different varieties of cotton seeds of different qualities, which can not be easily found by cotton farmers during trading and always causes great economic losses to cotton farmers and the downstream textile enterprises"(lines 45 - 52). "Therefore, accurate identification of cotton seed cultivars before cotton planting is very necessary, which is crucial for improving cotton quality, assisting cotton seed market administration, and safeguarding the interests of cotton farmers" (lines 55 - 58).

Potential limitations in practical applications: 1. Hyperspectral data acquisition equipment is expensive; 2. Model calculation is complex. There is still a long way to go from scientific research to practice, and relevant researchers need to work together to promote this cotton seed identification technology to the market as soon as possible.

What are the limitations of your work?

Response: There are still some limitations in this study. In this study, seeds were separately placed on the instrument for hyperspectral image acquisition. However, in actual cases, the seeds are stacked. Whether the model could still achieve good results in the case of stacking needs to be tested in future study. In addition, in this study, five cotton varieties widely planted in Xinjiang, China were studied. However, in the cotton seed trade market, there are many varieties available. So, in the follow-up research, more cotton varieties will be included in the study to improve the stability and applicability of the model. We have added these limitations in the Conclusion section. Please see lines 426 - 432. 

"However, there are still some limitations in this study. In this study, seeds were separated for hyperspectral image acquisition. However, in actual cases, the seeds are stacked. Whether the model could still achieve good results in the case of stacking needs to be tested in future study. In addition, in this study, five cotton cultivars widely planted in Xinjiang, China were studied. However, in the cotton seed trade market, there are many cultivars available. So, in the follow-up research, more cotton cultivars will be included in the study to improve the stability and applicability of the model".

Reviewer #2: The paper presents an experimental study for cotton seed identification using hyperspectral images analysis. The authors have tried various methods at each stage of the classification task, starting from the preprocessing stage of noise removal, feature selection and model training and evaluation. Overall, the paper is well-written with clearly explained steps.

However, I would like to suggest the following points which the authors can incorporate in their manuscript after revision.

1- The contributions need to be highlighted clearly in the introduction, with clear distinction from the state-of-the-art.

Response: Many thanks for your review and comments. We have added the contributions in the introduction according to your suggestion. Please see lines 90 - 101, 45 - 52 and 55 - 58.

"At present, most researches on the identification of seed cultivars by hyperspectral imaging technology focus on the seeds of corn, wheat, soybean, etc. (Wang et al., 2021; Zhao et al., 2022; Thomas et al., 2023). The performance of hyperspectral imaging technology in the identification of cotton seed cultivars is still unclear. In addition, previous hyperspectral data acquisitions require the use of a mobile platform, making the process cumbersome. Especially, the acquired hyperspectral images have to be corrected by manual computation (Huang et al., 2016; Jin et al., 2022). However, the portable Specim IQ handheld push-broom hyperspectral camera offers real-time data acquisition and ease of operation. Especially, the data acquired with the Specim IQ camera are already corrected, eliminating the need for additional computation, which improves efficiency. The usability of the Specim IQ camera was already demonstrated by Jan et al. (2018), who compared it with the Specim V10E sensor and evaluated its measurement quality. " (lines 90 - 101)

Practical contribution: "The market's high quality requirements for textile products and cotton require cotton seeds to reach a certain level of purity and quality. Xinjiang is China's largest cotton production base, and there are a large number of cotton varieties in the local seed markets. Driven by economic interests, some merchants often mix different varieties of cotton seeds of different qualities, which can not be easily found by cotton farmers during trading and always causes great economic losses to cotton farmers and the downstream textile enterprises." (lines 45 - 52)"Therefore, accurate identification of cotton seed cultivars before cotton planting is very necessary, which is crucial for improving cotton quality, assisting cotton seed market administration, and safeguarding the interests of cotton farmers". (lines 55 - 58)

The significance of this study is to propose a rapid, non-destructive, and accurate method to identify cotton seed varieties. This helps to improve cotton yield and quality, assist the seed market administration, and safeguard the interests of cotton farmers. Besides, this can also help to avoid the multi-variety cotton mixing processing-induced poor quality of the textile products, reducing the risk of economic losses of textile enterprises. 

References:

Michelon, Thomas B., Elisa Serra Negra Vieira, and Maristela Panobianco. "Spectral imaging and chemometrics applied at phenotyping in seed science studies: a systematic review." Seed Science Research (2023): 1-14. DOI10.1017/S0960258523000028.

Zhao, Liang, S. M. Haque, and Ruojing Wang. "Automated seed identification with computer vision: challenges and opportunities." Seed Science and Technology 50.2 (2022): 75-102. DOI10.15258/sst.2022.50.1.s.05.

Hong, Wang, et al. "Progress in research on rapid and non-destructive detection of seed quality based on spectroscopy and imaging technology." Spectroscopy and Spectral Analysis 41.1 (2021): 52-59. DOI10.3964/j.issn.1000-0593(2021)01-0052-08.

Jin SL, Zhang WD, Yang PF, Zheng Y, An JL, Zhang ZY, Qu PX, Pan XP. Spatial-spectral feature extraction of hyperspectral images for wheat seed identification. Comput. Electr. Eng. 2022, 101, 108077. DOI10.1016/j.compeleceng.2022.108077.

Huang M, He CJ, Zhu QB, Qin JW. Maize seed variety classification using the integration of spectral and image features combined with feature transformation based on hyperspectral imaging. Appl. Sci. 2016, 6, 183. DOI10.3390/app6060183.

Behmann J, Acebron K, Emin D, Bennertz S, Matsubara S, Thomas S, Bohnenkamp D, Kuska MT, Jussila J, Salo H.et al. Specim IQ: Evaluation of a new, miniaturized handheld hyperspectral camera and its application for plant phenotyping and disease detection. Sensors 2018, 18, 441. DOI10.3390/s18020441.

2- The literature survey is not thorough. It should highlight clear differences and drawbacks that this work overcomes. A citation of Sun et al. The authors should clearly mention how this work is different from theirs, or is it just the application to cotton seeds that is different.

Response: We have improved the Introduction section according to your comment. Especially, we have added the differences and drawbacks that this work overcomes. Please see lines 90 - 101.

"At present, most researches on the identification of seed cultivars by hyperspectral imaging technology focus on the seeds of corn, wheat, soybean, etc. (Wang et al., 2021; Zhao et al., 2022; Thomas et al., 2023). The performance of hyperspectral imaging technology in the identification of cotton seed cultivars is still unclear. In addition, previous hyperspectral data acquisitions require the use of a mobile platform, making the process cumbersome. Especially, the acquired hyperspectral images have to be corrected by manual computation (Huang et al., 2016; Jin et al., 2022). However, the portable Specim IQ handheld push-broom hyperspectral camera offers real-time data acquisition and ease of operation. Especially, the data acquired with the Specim IQ camera are already corrected, eliminating the need for additional computation, which improves efficiency. The usability of the Specim IQ camera was already demonstrated by Jan et al. (2018), who compared it with the Specim V10E sensor and evaluated its measurement quality.”

Sun et al.'s research focused on black bean seeds, while this study focused on cotton seeds; (2) Sun et al.'s research involved complex data collection, while the data collection in this study is simpler; (3) Sun et al.'s research did not involve data pre-preprocessing, while the data were preprocessed with different methods and the outcomes were compared in this study.

References: 

Michelon, Thomas B., Elisa Serra Negra Vieira, and Maristela Panobianco. "Spectral imaging and chemometrics applied at phenotyping in seed science studies: a systematic review." Seed Science Research (2023): 1-14. DOI10.1017/S0960258523000028.

Zhao, Liang, S. M. Haque, and Ruojing Wang. "Automated seed identification with computer vision: challenges and opportunities." Seed Science and Technology 50.2 (2022): 75-102. DOI10.15258/sst.2022.50.1.s.05.

Hong, Wang, et al. "Progress in research on rapid and non-destructive detection of seed quality based on spectroscopy and imaging technology." Spectroscopy and Spectral Analysis 41.1 (2021): 52-59. DOI10.3964/j.issn.1000-0593(2021)01-0052-08.

Jin SL, Zhang WD, Yang PF, Zheng Y, An JL, Zhang ZY, Qu PX, Pan XP. Spatial-spectral feature extraction of hyperspectral images for wheat seed identification. Comput. Electr. Eng. 2022, 101, 108077. DOI10.1016/j.compeleceng.2022.108077.

Huang M, He CJ, Zhu QB, Qin JW. Maize seed variety classification using the integration of spectral and image features combined with feature transformation based on hyperspectral imaging. Appl. Sci. 2016, 6, 183. DOI10.3390/app6060183.

Behmann J, Acebron K, Emin D, Bennertz S, Matsubara S, Thomas S, Bohnenkamp D, Kuska MT, Jussila J, Salo H.et al. Specim IQ: Evaluation of a new, miniaturized handheld hyperspectral camera and its application for plant phenotyping and disease detection. Sensors 2018, 18, 441. DOI10.3390/s18020441.

3- The overall results are not being compared with state-of-the art methods. The authors have mentioned different preprocessing/filtering methods used in different literature review papers, but the final results need to be compared. If they are better, then why they are better.

Response: We have added the comparison with the preprocessing methods used in other literature. Please see lines 373 - 380.

"Compared with the SG smoothing combined with normalization spectral preprocessing method used by Huang et al. [4] in the study of the cotton seed cultivar identification based on hyperspectral image technology, the SNV-FD preprocessing method used in this study can effectively reduce the noise of cotton seed spectral data. Especially, the identification accuracy of the PLS-DA model based on the spectral data preprocessed by SNV-FD in this study is 4% (training set) and 6% (test set) higher than that of the PLS-DA model based on the spectral data preprocessed by SG smoothing combined with normalization".

References: 

Huang DY. Study on Identification Method of Delinted Cottonseeds Varieties Based on Hyperspectral Image Technology. M.A. Thesis, Shihezi University, Shihezi, China, 2018. 

4- Training and evaluation methodology and sample size: The authors have taken a relatively low sample size with 60 samples per class. Moreover, the authors have mentioned that the samples were hand-picked to minimize the intra-class variations. This has resulted in nearly 100% accuracy, which ideally, should not be presented. A perfect accuracy generally, suggests that the sample size needs to be increased.

Response: Thanks for your comment. During conducting the experiment, we discussed the following questions: (1) whether the sample size of 60 per variety meets the study requirements; (2) whether the result with 100% accuracy is abnormal? 

By reviewing the literature, it was found that Kong et al. (2018) used 56, 56, 55, and 58 seed samples for rice variety Zhongzheyou No.1, Zhongzheyou No.5, Zhongzheyou No.8, and Zhongzheyou No.86, respectively to explore the seed variety identification accuracy by near-infrared hyperspectral imaging and multivariate data analysis. They found that the accuracy of the SIMCA, SVM, and RF models reached 100%. Zhang et al. (2015) used near-infrared spectroscopy to collect spectral data of four varieties of vegetable seeds (Qibaoqingcai, Lvlingwuyueman, Aiqishuzhouqing, and Lvlingwutacai) (60 samples for each variety), and processed the raw spectra by SNV, to establish an identification model. They found that the accuracy reached 100%. Zhang et al. (2013) used 32, 25, 32, and 32 seed samples for watermelon variety Xiaofang, Xiufang, Lifang and Zhemi No. 5, respectively to explore the seed variety identification accuracy by near-infrared hyperspectral image technology. They found that the accuracy of the optimal discriminant model was 100%. 

Based on the above literature, we completed this experiment. However, your professional comment made us aware of the shortcomings of this study, and we will increase the sample size for further trials in the future. 

References:

Kong, W., Zhang, C., Liu, F., Nie, P., He, Y.: Rice seed cultivar identification using near-infrared hyperspectral imaging and multivariate data analysis. Sensors (Basel Switz.) 13(7), 8916–8927 (2013). https://doi.org/10.3390/s130708916.

Zhang, G., Sun, J.

---

## [Decision Letter · Decision Letter 1]

22 Apr 2024

Cotton Seed Cultivar Identification Based on the Fusion of Spectral and Textural Features

PONE-D-23-36209R1

Dear Dr. Guo,

We’re pleased to inform you that your manuscript has been judged scientifically suitable for publication and will be formally accepted for publication once it meets all outstanding technical requirements.

Kind regards,

Narendra Khatri, Ph.D.

Academic Editor

PLOS ONE

Additional Editor Comments (optional):

The requested corrections were extensively accomplished by the authors. Now my recommendation is accepted without any further revision.

I wish authors a great success.

Reviewers' comments:

Reviewer's Responses to Questions

**Comments to the Author**

Reviewer #2: All comments have been addressed

---

## [Editor Report · Acceptance letter]

2 May 2024

PONE-D-23-36209R1 

PLOS ONE

Dear Dr. Guo, 

I'm pleased to inform you that your manuscript has been deemed suitable for publication in PLOS ONE. Congratulations! Your manuscript is now being handed over to our production team.

Kind regards, 

on behalf of

Dr. Narendra Khatri 

Academic Editor

PLOS ONE